


# Responses to severe weather warnings and affective decision-making

Philippe Weyrich[1], Anna Scolobig[2], Florian Walther[3], Anthony Patt[1]

[1] Climate Policy Group, Department of Environmental Systems Science, Swiss Federal Institute of Technology (ETH Zurich), 8092 Zurich, Switzerland

[2] Environmental Governance and Territorial Development Institute, University of Geneva, 1205 Geneva, Switzerland

[3] Wetter-Alarm, GVB Services AG, 3063 Ittigen, Switzerland

*Correspondence to*: Philippe Weyrich philippe.weyrich@usys.ethz.ch



**Abstract**
Informing people of an impending hazard can lead them to adopt behavior to mitigate the harm. In this study
we examine whether giving more information, and giving it earlier, leads to a greater behavioral response.
Our results, which are contextually dependent, show that providing more information has no effect on
behavior, and that longer lead times lead to less behavioral change. These results conflict with those from
previous studies. These previous studies differed from ours in terms of the research methods: while past
studies examined people's anticipated responses to hypothetical warnings, we conducted a field experiment
to observe people's responses to actual warnings of real hazards. Theory from cognitive science suggests that
this difference matters. In situations of high stress people may make decisions using a faster decision pathway
that is rather emotion-driven, while in less stressful situations they are more likely to base their decisions on
information. The difference between actual and hypothetical warnings would capture this mismatch in stress
levels, and account for the divergent findings. At the same time, the cognitive theory has been hard to test in
the field, because of the ethical challenge of submitting people to actually dangerous conditions. Therefore,
our results are not only relevant for the design of warning information, but also provide important empirical
support for the theory of different decision-making pathways.

**1.    Introduction**
To the extent that people make decisions based on information, it would seem right that the more information
they receive about a situation demanding potential action, and the earlier they receive it, the better they can
adjust their behavior. However, there is evidence that people often make decisions based on their emotional
response to information (Slovic et al., 2004, 2007). In such cases, more information is not necessarily better.
Moreover, which decision-making pathway people utilize may depend on the context. However, this pathway
is not necessarily self-exclusive and could involve the interaction of information-based reasoning and
emotions (Kahneman, 2011). Here, we investigate the effectiveness of different kinds of information, as well
as its timing, used in warning people about impending high impact weather events. Our primary focus is on
the difference between standard warnings (SW), which describe the weather event itself, compared to impact-
based warnings (IBW), which, in addition, describe the impacts that result from the weather.

Research in social sciences has broadly accepted two ideas about human nature. The analytical or cognitive
idea suggests that people make rational decisions based on formal logic, risk assessment and statistical
probabilities, for instance on the impacts and likelihood of a hazard (Loewenstein et al., 2001; Slovic et al.,
2004). This system is rather slow as it requires mental work, which is effortful and orderly (Kahneman, 2011;
Slovic et al., 2004). The affective decision-making relates to the importance of emotions and feelings in
making decisions (Slovic et al., 2004). It operates automatically and fast, with neither effort nor sense of
voluntary control, although it is often influenced by beliefs, or mental models, about how the world works
(Morgan et al., 2002; Slovic et al., 2004).

Research that has investigated whether feelings, information-based action or both influence people's
behaviors related to risks has primarily relied on laboratory studies. For example, scholars have used different
messages to manipulate affect by increasing or decreasing perceived benefits and risks of different
technologies (Finucane et al., 2000). In two experiments, these researchers demonstrated that affect
influenced judgments directly and was not simply a response to a prior deliberate evaluation. In only two
studies, which we describe below, have researchers evaluated behavior under varying conditions of actual
fear, something that cannot be simulated in a laboratory.

A real-world situation where the emotional decision-making pathway could dominate is the response to
warnings of potentially life-threatening weather events, such as tornados or severe storms. Research that is
based on information-based decision-making has suggested that message content and style are important
factors in determining whether people take self-protective behavior to an extent that rational analysis would
deem appropriate (Mileti and Sorensen, 1990). In order to be effective at inducing such behavior, a message
should contain five information elements – hazard, location, time, guidance and source – which should be
addressed each by five stylistic dimensions – specificity, consistency, accuracy, certainty, and clarity (Mileti
and Sorensen, 1990). A warning with these characteristics is easy to understand, to believe and to personalize
for the recipient, identified as prerequisites for triggering behavioral change (Mileti and Peek, 2000). Thus





IBW, which provides more specific and clear information on the impacts of the hazard, should help people
to better understand the message compared to SW. IBW should also increase the personalization of risk and
make people feel more concerned for their safety, resulting in stronger behavioral response compared to SW.
For example, some people have difficulties to interpret a "heavy" rainfall warning, indicating 100 mm of
rain, into effective impacts. In this case communicating specific impacts, for instance on road and rail
transport, and possibilities of delays, ought to improve warning effectiveness. Interviews with forecasters,
emergency managers and broadcast meteorologists (Harrison et al., 2014; Losego et al., 2013), as well as
with officials from the public and private sector (Weyrich et al., 2018) all reveal a widespread belief within
the expert community that providing impact information creates an added value in the specific case of high
impact weather warnings.
Recent studies offer empirical support for this belief, although the results are somewhat mixed (Kox et al.,
2018). For example, scholars showed that IBW, compared to SW, positively influenced the recipient's sense
of threat and concern associated with a hypothetical event, as well as their understanding of the potential
impacts (Morss et al., 2018; Potter et al., 2018; Weyrich et al., 2018). More importantly, the IBW of the
hypothetical event resulted in a greater likelihood of people planning to take self-protective action, should
such an event occur  (Casteel, 2016; Morss et al., 2018; Weyrich et al., 2018). There have also been
contradictory findings. One study detected no effect of IBW on perception of warning credibility or on
intended behavioral response (Perreault et al., 2014), while another study identified a threshold beyond which
increasing the projected impact of a storm no longer significantly increased the probability of taking
protective action (Ripberger et al., 2014). All of these empirical studies, however, share a common research
design: they used hypothetical scenarios, and relied on people's anticipated and intended reactions to study
the effects of IBW. For example, in one study of tornado warnings, the effectiveness of IBW was examined
with respondents in the hypothetical role of a factory operator having to decide whether to order workers to
take shelter in response to SW and IBW (Casteel, 2016). In another study, participants had to imagine that
they would be hiking in the Swiss mountains when receiving a thunderstorm warning, and then had to decide
upon several intended actions; those receiving an IBW were more likely to modify their plans than those
receiving an SW (Weyrich et al., 2018).
If indeed it is feelings that dominate behavioral decision-making in real-life situations, then it may be that
these studies on the effectiveness of IBW are poor predictors of actual behavior, as it is unlikely that the
respondents experienced real feelings of fear, since they were not actually at risk. Two studies exist that have
looked at actual self-protective behavior during a crisis suggest this to be the case. Researchers in Indonesia
investigated evacuation behaviors and intentions during tsunamis, and observed that feelings, and not rational
evaluation, drive decision-making (McCaughey et al., submitted). Their findings suggest that under an
imminent threat of life, information-based action may be absent or far less influential than feelings. Scholars
from the Netherlands analyzed the behavioral effects to mobile fire warning messages (Gutteling et al., 2017).
They found that emotions and the social environment were the main predictors for adaptive behavior. Even
though perceived message quality was significant, other factors, such as perceived threat, were insignificant.
These results confirm the importance of affective reactions as a driver for behavior.
If affective decision-making is the dominant pathway in real-world crises, then SW may provide all the
information that is needed to trigger the feelings of fear, with IBW adding no additional trigger. We speculate
that hazard severity and warning lead time could also influence the response to weather warnings in different
ways depending on the model of decision-making. If information-based action dominates then more severe
events and greater lead times should generate a greater behavioral response: longer lead times would translate
into greater ease of preparing for and actually taking self-protective behavior. If affective decision-making
dominates, however, more severe events and shorter lead times should increase response, since the fear
should be heightened at the time of the information reception.
There have been two studies examining the effect of warning lead time, and one related to event magnitude.
One of these examined tornado response, and showed that an increase in lead time up to about 15 minutes
reduces fatalities, while lead times longer than 15 minutes increase fatalities compared with no warning
(Simmons and Sutter, 2008). The second study showed more generally that people have lead time preferences
that do not always match with what the warning system offers, and that they engage in different protective
behaviors depending on the lead time (Hoekstra et al., 2011). The one study examining event magnitude,





using a hypothetical survey design, showed that the greater the severity the more likely people are to take
protective action (Kox and Thieken, 2017). Perceived severity of the hazard is also used in many decision-
making theories. For instance in Protection Motivation Theory, it is one of four core perceptions that form
the basis for decisions about how to respond to a threat (Maddux and Rogers, 1983).
In this paper we report on results from a randomized control trial in which we disseminated wind warnings
through an existing smartphone application of a Swiss weather provider (*Wetter-Alarm*), and collected real-
time data on people's responses. The information that people received varied randomly in terms of being SW
or IBW, and given that there were a number of events for which the warnings were issued, in terms of both
the warning lead time and the events' anticipated severity.
**2. Materials and Methods**
The method used here was a large field experiment conducted in Switzerland, which tested for effects of
warning type, severity level and lead time on warning response. SW and IBW for wind were disseminated to
users via the smartphone weather application (app) 'Wetter-Alarm'. The application resulted out of a
cooperation between the GVB Services AG (which is responsible for the app) and SRF Meteo, which
provides the weather (i.e., warnings for frost, thunderstorm, slipperiness, rain, snow and wind among others).
The users could receive warnings for three severity levels: moderate (slight risk of damage), severe (increased
risk of damage) and very severe (big risk of damage or even risk of death). The standard warnings
disseminated in the Wetter-Alarm app included information about the type of hazard, its severity, the timing
and location, as well as some general behavioral recommendations (e.g., secure lose items or avoid forests).
Figure 1 shows a standard wind warning of medium severity. The impact-based warning included the
identical information than the SW, but with an additional impact information of the weather which are shown
in Table 1. We developed these messages based on publicly available information on impacts of wind in
Switzerland and in close collaboration with the staff of Wetter-Alarm. A link was provided at the end of the
warning message, which directed participants to a short questionnaire. The questionnaire was available from
the moment on when the warning message was disseminated until the end of the event. We focused on severe
wind due to its frequency, the time of the year (winter season) and the possibility to investigate different lead
times. We collected data for two wind severity levels: moderate and severe. As this research involves research
on humans, appropriate ethical procedures were followed, which was approved by the Ethics Commission of
ETH Zürich. Participants voluntarily participated once they had been informed about the research project and
signed a declaration of consent. They received no incentive to complete the survey.
Figure 1. Standard wind warning of medium severity level for the region La Côte/Morges.
Table 1. Additional impact-based information per severity level in the impact-based warnings.
A total of 3,223 participants completed the online survey from 1.12.2018 to 10.02.2019. We excluded 611
people from the analysis as they believed to have responded to a warning message with a different severity
level than it was actually the case. This can be explained by the fact that the warning message they received
initially, was updated in the meantime (e.g., from a moderate to a severe level) or that the participants received
multiple warning messages for different locations and got confused. Thus, to avoid any possible
misinterpretation of data, the analysis was conducted with data from 2,615 participants that indicated the
correct severity level. As respondents were randomly assigned to either a SW or IBW, the subgroups are
roughly even (1,364 and 1,247). However, more people responded to severe warning messages (n=1,667)
than to moderate messages (n=948). No very severe wind was observed. Warning lead times also differed
and people were grouped into three groups depending on when they looked at the warning message (i.e.,
participated in the survey); during the wind event itself (n=932, 35.6 %), in the 6 hours preceding the wind
event (17.1 %, n=448) and prior to 6 hours, 47.2 % (n=1,235). On average, people responded to the survey
5.14 hours in advance of the wind event.
Information about the basic socio-demographic characteristics of the sample is provided in Table 2. The
sample matches the profile of the general Wetter-Alarm app user which is older (48.8 years) than the Swiss
average (43.14 years), is more often male (63.1 %) than female (Swiss average 49.5 % vs. 50.5 %) (FSO,
2017b) and slightly more educated than the Swiss population (FSO, 2017a). As the survey was conducted


online based on actively users of the app Wetter-Alarm, it did not reach people who did not download the application, who do not actively use the app or who do not have internet access. People could only participate once in the survey, which was guaranteed through posing the question whether they already participated in a Wetter-Alarm survey recently.

Table 2. Socio-demographic characteristics of participants in the field experiment.

In the survey, we asked questions on warning perception and understanding. Perceptions that we measured using a five-point Likert scale from 'totally disagree' to 'totally agree' were credibility, and concern. We measured three types of understanding: the warning, the threats to safety, and how to respond. Then, we asked participants whether the weather described in the warning would pose a risk to them and whether it would affect them in carrying out their usual activities (e.g., commuting, working, shopping etc.). If they answered yes, they continued with the survey. The following three questions were used to build the variable behavioral response. First, participants had to indicate whether they responded to the warning. If answered 'Yes', they had to indicate whether they adapted, but continued with their activities or whether they cancelled their activities (respectively taking other measures for protection). If answered 'No', participants had to indicate whether they would not change their behavior, or still plan to do so, i.e., adapting activities or cancelling activities. Thus, we computed the variable behavioral response on a five-point scale (1= no action planned, 2= plan to adapt, 3= plan to protect, 4= did adapt, 5= did protect). We used this scale from no response to strongest risk minimizing behavior as we believe that it catches more variance than only the binary question on whether people responded to the warning or not. Similar to other research (Gutteling et al., 2017), we used a battery to ask what kind of feelings the warning did trigger: relaxed, anxious, concerned, reassured and angry (five-point Likert-scale from 'not at all' to 'very much'). These questions were used in other studies that investigated behavioral responses to emergencies (Gutteling et al., 2017; Kievik et al., 2012; Kievik and Gutteling, 2011) and thus seemed to be an appropriate measure also in this study's context. The items 'relaxed' and 'reassured' were inverted and the scale yielded good internal consistency (Cronbach's alpha = 0.68, N = 5). We also gathered data on whether people consulted other information for advice or confirmation (binary question Yes/No). Finally, we collected information on the most important personal factors: gender, age, and education. The full questionnaire is available in the supplement.

For the data analysis we use standard statistical software (IBM SPSS 25) to conduct a factorial ANOVA to study the effects of warning type, severity level and lead time on behavioral response. In addition, we did a multiple regression analysis to investigate the effects of other covariables (e.g. warning perception and understanding) on behavior.

## 3. Results

We first describe the effects of warning type, lead time, and event magnitude on participants' perception and understanding. We summarize the mean values in the appendix. IBW were not perceived to be more credible, nor to be better understood in terms of the warning, the threats to safety, and how to respond compared to SW. People were only slightly more concerned for their safety when receiving IBW. Participants' perception and understanding did not change with different lead times. However, people indicated higher perceived concern levels for severe compared to moderate warnings. Not surprisingly, people reported increased feelings with decreasing lead times and increasing severity levels.

To analyze the effects of warning type, severity level and lead time on behavior, we focus on those people who indicated the warning to be relevant and analyzed their behavior. Fifty-four percent of people (n=1426) reported that the warning message affected their personal safety, impacted their daily routine or both. The majority of those people already changed their behavior, either by adapting their activities (35.2 %) or by cancelling them (25.7 %). Fewer people indicated that they still planned to adapt (22.7 %) or to cancel (6.9 %) their activities. Nine percent of people reported not changing their behavior, even though the message was found to be relevant. We conducted a factorial ANOVA (2 (Warning Type) X 3 (Lead Time) X 2 (Warning Severity Level) predicting behavior, which showed no effect of warning type (p=.963), but effects of lead time ($F_{(1, 1410)}=11.00$, $p<.001$, $\eta_{p2}=0.02$), and of severity level ($F_{(1,1410)}=12.21$, $p<.001$, $\eta_{p2}=0.01$).




The Bonferroni post hoc test revealed that changing behavior was significantly lower for long lead times
compared to short (p=.007) or no lead times (p<.001). All interaction effects between any of the three
variables (type, severity and time) on behavior were non-significant (p-values between 0.360 and 0.546).
Figure 2 underlines that IBW did not result in greater behavioral response compared to SW. However, as Fig.
3 highlights, lead time and warning severity significantly influenced people's decisions to change behavior:
decreasing lead times and increasing severity level resulted in a greater response. We also observe that the
differences in behavioral response between moderate and severe warnings are quite low for long lead times.
This difference becomes more important for shorter lead times. However, the interaction is not significant
(p=.360). In the next set of relationships, we examined what additional factors influence behavioral response.
In specific, we analyze the relationship between feelings, respectively warning perception/understanding,
and behavioral action. Table 3 shows that irrespective of warning type received, feelings (a unit increase in
feelings 0.25 unit increase in changing behavior), perceptions of credibility ($\beta=0.134$) and concern ($\beta=0.098$),
as well as understanding the threats ($\beta=0.193$) and how to respond to the message ($\beta=0.154$) significantly
influence taking protective action. Moreover, age ($\beta=0.081$) and information behavior ($\beta=0.100$) showed
significant positive effects. Thus, the more people felt in danger, the better they perceived or understood the
message, the older they are, and the more they looked for information, the more likely they were to undertake
strong risk minimizing behaviors. The linear regression analysis again confirms the importance of lead time
($p<.001$) and warning severity ($p<.01$) on the behavior variable. With decreasing lead times, people are more
likely to take protective action (a unit increase in lead time predicted a 0.154 unit decrease in changing
behavior). For severe warnings, people were also more likely to change their behavior (by 0.073 unit)
compared to people who received moderate severity warnings.
Figure 2. Mean likelihood to change behavior for the two warning types (SW and IBW) and the three lead times (no,
short and long), respectively the two severity levels (moderate and severe).
Figure 3. Mean likelihood to change behavior for all three lead times (no, short and long) and two severity levels
(moderate and severe).
Table 3. Multiple linear regression with the behavioral change as dependent variable.
4. **Discussion**
This research investigates the effectiveness of impact information, as well as its timing, used in warning
people about an imminent threat. Our results show that while IBW result in no greater behavioral response,
decreasing lead times and stronger severity level do increase response. Taken together, these results suggest
that affective decision-making appears to be the dominant mode of decision-making in real-world situations.
IBW do not significantly impact warning perception and understanding, nor do they result in greater
behavioral response compared to SW. This result contradicts the majority of previous studies that used
hypothetical situations to collect their data (Casteel, 2016; Morss et al., 2018; Weyrich et al., 2018). We
speculate that this difference in research findings can be explained by the different levels of fear experienced
in a hypothetical and a real crisis. Unlike in an imagined situation, where information-based action is the
dominant factor, our findings suggest that in a crisis situation, real feelings of fear arise and dominate
decision-making. We assume that SW provide all the information that is needed to trigger the feeling of fear.
Indeed, IBW may leave less to the imagination of the recipient, which could – in some cases – dampen the
fear response.
Our results on the effects of lead time and hazard severity are also consistent with affective reactions. We
observe lead time and self-protective behavior to be inversely correlated and find that increasing lead times,
decrease the likelihood to engage in greater behavioral response and observe the greatest response when the
event has already unfolded. These results complement other research on different lead times for tornado
warnings (Hoekstra et al., 2011). We also show that stronger events generate a greater response than weaker
events, which is in line with previous research (Kox and Thieken, 2017). Moreover, we observe that longer
lead times do not generate a greater additional response to stronger rather than weaker events. This interaction
(even though not significant) is in line with an affective reaction: with long lead times, the additional fear
associated with the stronger event may dissipate, meaning that the stronger events would generate little more
response than weaker events.




These findings support scholars who reached a similar conclusion when investigating evacuation behaviors following a strong earthquake (McCaughey et al., submitted). Nonetheless, cognitive factors, such as warning perception and understanding can also influence decision-making. In our study, four of these information-based attributes correlate with changing behavior and thus seem to be obvious prerequisites for behavior (Gutteling et al., 2017). Indeed, the two decision-making pathways should not be seen as independent systems, they can interact and influence each other as the rational process can modify, to some extent, the way we make intuitive and affective decisions by changing the normally automatic functions of attention and memory (Kahneman, 2011). The research also shows that the two systems are not always self-exclusive, for instance when people are asked to judge risk, they first consider how they feel about the risk and then collect further information, usually to support their feelings (Slovic et al., 2004). Therefore, further empirical studies of real-world crises are needed to understand if, and how feelings and information-based action interact to influence people's behaviors to risks.

### 5. Conclusions

We conclude that practitioners cannot assume that additional impact-based information necessarily results in greater behavioral response in real-world crises. Appropriate lead times and a communication that addresses the decision-makers' feelings (e.g., by relying on images) may be more beneficial and result in a stronger behavioral response. Ultimately, the results show that people may respond differently in a field than in a scenario-based experiment, based on more affective, respectively rational decision-making. This has serious implications for future research emphasizing that we should examine responses to risks using research designs that capture realistic conditions and be cautious in interpreting results from hypothetical research designs as these could be a poor predictor of actual behavior.

The research has some limitations. One shortcoming of this study is the absence of a very severe wind event in the winter season 2018/19 in Switzerland, and additional data should be collected for these events too. Indeed, most of the research on IBW used hypothetical warning messages of the most severe category, as people are least familiar with these messages and, thus, the added information could help them in decision-making. In consequence, the difference in the results on the effectiveness of IBW in our and previous studies could also be due - to some extent - to the differences in event severity level. Moreover, participants were self-selected as they had downloaded the weather app and decided whether or not to participate in the survey. This may indicate higher levels of weather awareness and knowledge, which could also be another explanation for the lack of effect of warning type. Another limitation is that, even though we collect data on actual behaviors in response to real-life warnings, these were still self-reported. Thus, additional research could analyze whether these results are valid for other natural hazards as well.

### Appendix

Table A. **Descriptive statistics**. M=Mean, SD= Standard deviation. Variables were measured on a five-point scale from 1='totally disagree' to 5='totally agree'.

|  | Warning type | | Lead time | | | Severity level | |
|---|---|---|---|---|---|---|---|
|  | SW | IBW | No | Short | Long | Moderate | Severe |
| Credibility perception | (M=4.07, SE=0.02) | (M=4.04, SE=0.02) | (M=4.09, SE=0.03) | (M=4.03, SE=0.04) | (M=4.04, SE=0.02) | (M=4.02, SE=0.03) | (M=4.07, SE=0.02) |
| Concern perception | (M=2.00, SE=0.03) | (M=2.10, SE=0.03) | (M=2.09, SE=0.03) | (M=2.02, SE=0.04) | (M=2.02, SE=0.03) | (M=1.91, SE=0.03) | (M=2.12, SE=0.02) |
| Understanding the warning | (M=4.38, SE=0.02) | (M=4.36, SE=0.02) | (M=4.36, SE=0.02) | (M=4.36, SE=0.03) | (M=4.39, SE=0.02) | (M=4.37, SE=0.02) | (M=4.38, SE=0.02) |
| Understanding the threat | (M=3.96, SE=0.02) | (M=4.02, SE=0.02) | (M=4.00, SE=0.02) | (M=4.02, SE=0.04) | (M=3.98, SE=0.02) | (M=3.99, SE=0.03) | (M=4.00, SE=0.02) |
| Understanding how to respond | (M=3.94, SE=0.02) | (M=3.96, SE=0.02) | (M=3.92, SE=0.03) | (M=3.98, SE=0.04) | (M=3.96, SE=0.02) | (M=3.94, SE=0.03) | (M=3.95, SE=0.02) |
| Affective reaction | (M=2.10, SE=0.02) | (M=2.15, SE=0.02) | (M=2.20, SE=0.02) | (M=2.13, SE=0.03 | (M=2.10, SE=0.02) | (M=1.98, SE=0.02) | (M=2.21, SE=0.02) |


**Author contribution**
PW designed and performed the research. PW analyzed the data and wrote most of the paper. AS helped designing the survey and FW performing the research. AP helped designing the research and structuring, respectively writing the paper.

**Data availability**
In the research design that we originally submitted to our Ethical Commission (equivalent to an Internal Review Board), we had stated that all data would be deleted from ETH computers after the end of the project, but would be stored on servers at our partner (Wetteralarm), and would be potentially used to improve the design of their mobile application. Thus, interested researchers should contact us, and we should be able to work with Wetteralarm to provide the data requested.

**Competing interests**
The authors declare that they have no conflict of interest.

**Acknowledgments**
We thank the entire Wetter-Alarm Team for their help in the implementation of our research design. Conducting the field experiment would not have been possible without their permission to use the Wetter-Alarm mobile application to. Within the Wetter-Alarm Team, we would like to especially thank Marlène Käsermann for her valuable inputs.

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

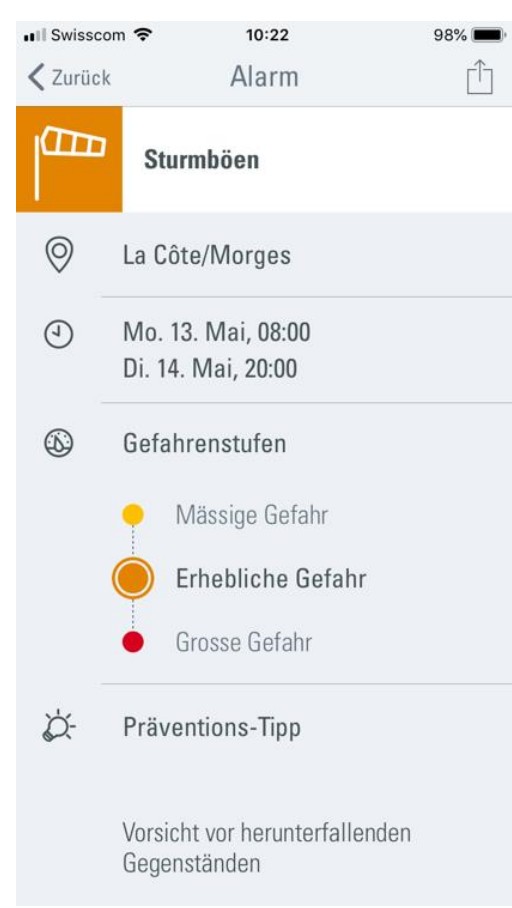

**Figure 1. Standard wind warning of medium severity level for the region La Côte/Morges.**




1  **Table 1. Additional impact-based information per severity level in the impact-based warnings.** Note that we did
2  not observe any very severe (level 3) warnings during the data collection period.

| Warning severity level | | |
|---|---|---|
| *Moderate (level 1)* | *Severe (level 2)* | *Very severe (level 3)* |
| Traffic delay | Traffic disruption or restriction | Traffic disruptions or standstill |
| Overturning of objects | Damage to individual buildings/roofs | Damage to buildings/roofs |
| Falling of smaller branches | Falling of big branches | Falling trees |



**Table 2. Socio-demographic characteristics of participants in the field experiment.**

| | |
|---|---|
| Gender | Males: n=1645, 63.1 %<br>Females: n=970, 56.9 % |
| Age | 48.8 years |
| Completed educational level | 34.6 % vocational school, 20.2 % university degree, 19.2 % collage, 18.9 % technical or high school, 7.1 % some compulsory education |



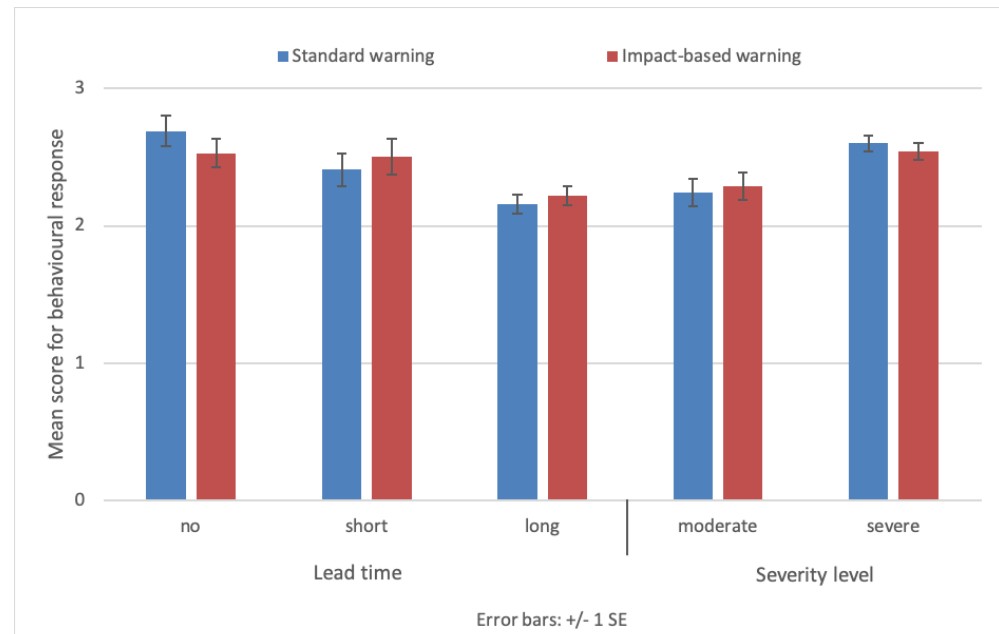

**Figure 2**. **Mean likelihood to change behavior for the two warning types (SW and IBW) and the three lead times**
**(no, short and long), respectively the two severity levels (moderate and severe).** Behavioral response was measured
on a five-point scale from no response to strongest risk minimizing behavior. For lead times, "no" indicates that
respondents considered the warning during the event, "short" refers to 0-6 hours prior to the event and "long" to more
than 6 hours. Error bars indicate +/- 1 the standard error. N=1426.

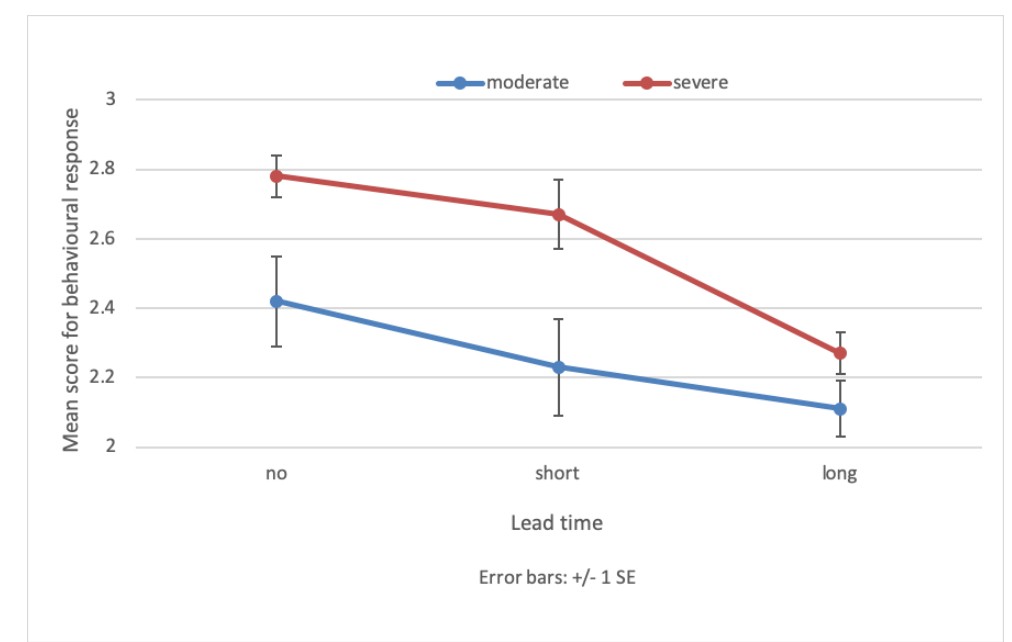

**Figure 3**. **Mean likelihood to change behavior for all three lead times (no, short and long) and two severity levels**
**(moderate and severe).** Behavioral response was measured on a five-point scale from no response to strongest risk
minimizing behavior. For lead times, "no" indicates that respondents considered the warning during the event, "short"
refers to 0-6 hours prior to the event and "long" to more than 6 hours. Error bars indicate +/- 1 the standard error. N=1426.



2 **Table 3. Multiple linear regression with the behavioral change as dependent variable.** $R^2$=0.36 (p<.001), N=1426.
3 Note: **p<.01, ***p<.001. Significant results are in bold. B indicates the unstandardized coefficients, SE the standard
4 error and β the standardized coefficients.

| | B | SE B | β |
|---|---|---|---|
| Constant | -.600 | .399 | |
| Gender (female=0; male=1) | -.107 | .073 | -.040 |
| **Age** (year scale) | **.008** | **.003** | **.081**** |
| Education level (scale 1-6) | .024 | .019 | .032 |
| **Credibility perception** (1-5 scale) | **.134** | **.055** | **.068*** |
| **Concern perception** (1-5 scale) | **.098** | **.041** | **.071*** |
| Understanding the warning (1-5 scale) | -.28 | 0.59 | -0.14 |
| **Understanding the threat** (1-5 scale) | **.193** | **.058** | **.108**** |
| **Understanding how to respond** (1-5 scale) | **.154** | **.053** | **.092**** |
| **Feelings** (1-5 scale) | **.250** | **.062** | **.122***** |
| **Information behavior** (no=0, yes=1) | **.690** | **.179** | **.100***** |
| **Lead time** (none=0, short=1, long=2) | **-.224** | **.038** | **-.154***** |
| **Warning severity level** (moderate=0, severe =1) | **.212** | **.077** | **.072**** |

