# Peer review of "Responses to severe weather warnings and affective decision-making"

_Natural Hazards and Earth System Sciences, 2020_

## Referee Comment (RC1) · Rainer Kaltenberger (Referee) · 18 Jun 2020

The paper presents the results of a Swiss field experiment to observe people's responses to severe weather warnings. In a randomized control trial over roughly 2.5 months, about 3,000 users received warnings for the hazard wind(storm) in two possible warning types via the smartphone weather application "Wetteralarm" and were requested to fill an online survey attached to each warning and targeting their behavioral response. One warning type, what the authors call "Standard Warnings (SW)", consisted on information about the (weather) hazard, severity level (three levels), timing, location and "some general behavioral recommendations (e.g. secure lose (sic) items or avoid forests"). The warning type "Impact-based warnings IBW" contained, in addition to the contents of a standard warning, a brief, rather general description of an

expected impact scenario. The contextually dependent results were summarized as "IBW did not result in greater behavioral response compared to SW", however, "lead time and warning severity significantely influenced people's decisions to change behaviour".

The overall presentation is well structured, easy to read and clear to understand by a wide and diversified audience. It would be of great benefit, if the authors would describe the "general behavioral recommendations" which are part of their SW, in more detail; e.g. by listing them in an additional table similar to Table 1 ("Additional impact-based information"). The term SW is chosen a little bit unfortunate, as in the European context standard meteorological warnings usually just include location, timing, hazard-type, severity level and eventually some meteorological information (e.g. Rainfall with amounts up to 100 mm), but generally no (generic) behavioral recommendations. In the Sendai context, behavioral recommendations are often seen, together with the impact description, already as a part of an IbW or Impact-oriented warning (e.g. [1]). Obviously some of the additional impact descriptions according to Table 1 (e.g. "Falling of smaller branches") are very similar to the behavioral recommendation ("Präventions-Tipp") of the depicted SW in Fig. 1, saying to be aware of falling items ("Vorsicht vor herunterfallenden Gegenständen"). This provision of rather little additional information in the warning message might be one of the main reasons for the principal finding, that in their field experiment IBW did not result in greater behavioral response compared to SW which should be discussed by the authors. It is likely, that SW without behavioral advices, IBWs with richer (or more empathic) impact descriptions or generally more tailormade warning texts would have changed the results significantly. Acknowledging this and other limitations, e.g. just warnings for one hazard were investigated, no warnings with the highest severity level due to the relatively short period of just 2.5 months, some concluding statements would benefit from being expressed a little bit more cautious and less generalized.

Although having some limitations, this paper describes a good example for how field

experiments targeting the measuring and understanding of the behavioural response to meteorological warnings should be designed and carried out. It provides interesting information for weather services, civil protection agencies and the research community. The introduction acts as a well-written resource, providing both social scientists and meteorologists a comprehensive overview in the highly interdisciplinary and dynamically evolving field of warning research.

[1] Kaltenberger, R., Schaffhauser, A., and Staudinger, M.: "What the weather will do" – results of a survey on impact-oriented and impact-based warnings in European NMHSs, Adv. Sci. Res., 17, 29–38, https://doi.org/10.5194/asr-17-29-2020, 2020.

Rainer Kaltenberger, ZAMG/EUMETNET Meteoalarm

---

## Referee Comment (RC2) · Anonymous Referee #2 · 8 Jul 2020

Dear editorial team, please accept my apologies that I can not provide another review of this manuscript due to my contract ending in Leeds on July 10th. Please find my comments below on this, overall, very good paper. Thank you for choosing me as a reviewer. Kind regards, Astrid Kause

GENERAL COMMENT The study represents an important and timely contribution to communications of weather warnings. It contributes to understanding the effectiveness of different weather warning types and behavioural responses to these warnings. The non-hypothetical 'field study'-approach is highly valuable when trying to understand responses to severe weather events which potentially cause high costs, and threaten lives. Also, the manuscript stands out because it is very easy to follow.

SPECIFIC COMMENTS INTRODUCTION. #1 In the introduction, I appreciate that the

authors try to draw on relevant theory from psychology and decision sciences. However, how the introduction stands now, it is long and in some parts confusing. I'd recommend to focus more on the applied value of this applied study and the factors varied in them, and what it adds to the current literature. The cited work of Casteel (2016, for example) or Andrea Taylor, University of Leeds, UK (2019) are excellent examples for how this could be done. If it is necessary to draw on several theories from cognitive and decision sciences, please clarify what are the precise predictions regarding the main independent variables explored here, based on these theories. This should be either based on theories, previous findings, or ideally integrate both. Regarding theory, I'd suggest narrow the theoretical focus and delineate precise predictions; and then pick these up again in the discussion and discuss whether and how they have been met, and why. If however, the aim of the authors is to review several theories rather than focusing on only one, please clarify why and how those were selected and relevant for IBW (for example construal level theory would then currently be missing).

**2 On a side note, I'd like to put forward that a dual systems-approach is widely accepted is not correct – the 'ecological rationality' approach (see reference to Gigerenzer & Gaissmaier, 2011 below) is an entire interdisciplinary research paradigm in psychology and decision sciences (in social sciences more broadly, a much wider range of theories exist). It contrasts the dual system-approach. In my view, the inherent problem of the latter is that it doesn't clarify, generally and also in potentially threatening situations where people may rely on weather warnings, what a 'rational' and therefore 'correct' decision actually is. Extreme weather leads to situations characterised by high uncertainty, where not all information is known and a 'rational' decision, based on all available information, is impossible to make. Also, quick and intuitive decisions could potentially be very helpful and adaptive in such uncertain emergency situations, if performed in the right decision context. While the introduction of the current paper is not the place to reflect this entire debate, the point I wish to illustrate is that the most important existing theories and predictions based on these need to be selected and described more carefully. #3 Having a more clear outline in the introduction would**

also clarify from an applied viewpoint what type of behavioural response is actually adequate in the authors' view (and 'rational', though I'd recommend to skip this term overall) – this may be based on general recommendations by weather warning services, or insurances, or Table 2? If there is an adequate response, please describe it in the introduction. If not, that also needs to be clear, because then there isn't such a thing as a rational decision in this context. A more applied focus as mentioned above would help in clarifying this.

**4 This would also help re-structuring the abstract according to short background, research question, design, method and sample, main findings and a short comment on results. Currently, it doesn't precisely reflect what was done here – and lacks for example the impact-based weather warning manipulation, or introduces the term 'stress' which doesn't re-appear prominently anywhere else (as far as I can see).**

METHODS. #5 P.5 l.10 says "we asked participants whether the weather described in the warning would pose a risk to them and whether it would affect them in carrying out their usual activities (e.g., commuting, working, shopping etc.). If they answered yes, they continued with the survey" – wouldn't this exclusion of people who don't change, be an explanation of why no effects were observed – if for example more people didn't react to a SW, compared to a IBW? #6 Page 5 l.34 reports that the regression is conducted on behaviour, but the regression table header indicates that it predicts behaviour 'change' – please clarify whether it predicted absolute behaviors or a difference score reflecting change (I don't think it did the latter). #7 Table 2: Please report a measure of dispersion for age, and a range. #8 Figure 2: I think this is not about likelihood (as the header implies) but a continuous score measuring actual behavior. I think the figure header should accordingly be something like "Mean self-reported behaviour in response to two warning types, ...". Please avoid the use of acronyms or explain them again in the caption to allow the figure to stand on its' own. I'd also prefer to see the full scale that was presented to participants on the y-axis; to get an idea about effect magnitude. Please do also apply these points to all other figures where

appropriate. #9 Table 3 and overall results section: The use of standard significance levels and \*\*\* has been widely criticized in psychology and decision sciences, and other fields (see for example https://www.nature.com/articles/d41586-019-00857-9 or https://journals.sagepub.com/doi/abs/10.1177/0959354302126005 for some context). Albeit still widely in use unfortunately, I'd thus refrain from marking results with \*\*\* and in bold, dependent on these levels; and follow reporting standards outlined in this literature; including for example confidence intervals where appropriate for all study results.

DISCUSSION. #10 As the authors note in line 27 on page 7, the study is based on self-reports and a self-selected sample. Please acknowledge the dearth of literature on samples and self-reports in psychology and social sciences more generally where this has been criticized, and provide a recommendation for future studies.

TECHNICAL CORRECTIONS #11 Please make sure that the wording is consistent – for example, 'survey' and 'questionnaire' are used interchangeably #12 Table A: Reports standard errors, not standard deviations. Please adjust the acronym in the header #13 P.5 l. 10 – this is actually subjective understanding – to measure actual understanding, you would conduct a test on recall or accurate reproduction of the communicated information, which I don't think has been done here. #14 I'd recommend to avoid somewhat jargon'y words, such as decision-making pathways. Not everyone in an interdisciplinary audience may understand what those are. Acronyms (and translations to understand what those mean) are not introduced everywhere; such as GVB Services AG on page 4. #15 I appreciate the comparison to the general population. Consider restructuring Table 2 and include the comparison to the Swiss population which is currently in the text on p. 4, l. 51, so that it allows more easily comparing the sample to the general population; this also avoids redundancy between text and table. #16 Table 3: rather than reporting for each predictor on which scale it has been measured, add a note and describe groups of factors that have been measured on 1-5 scales, 1-3 scales, and binary ones. #17 Some references use capitals for each title word, others don't (for example Potter et al. 2018).

Reference Gigerenzer, G. & Gaissmaier, W. Heuristic decision making. Annu. Rev. Psychol. 62, 451–82 (2011).

---

## Author Response (AR1)

**Point-by-point response to the reviews**

The manuscript has been revised according to the comments and suggestions received by the two reviewers. We would like to thank them for their inputs and useful insight. The reviewers' comments are reproduced below, followed by our response.

Reviewer 1

Comment part 1: The overall presentation is well structured, easy to read and clear to understand bya wide and diversified audience. It would be of great benefit, if the authors woulddescribe the "general behavioral recommendations" which are part of their SW, in moredetail; e.g. by listing them in an additional table similar to Table 1 ("Additional impact-based information").

Response: We thank the Reviewer very much for the comment. The weather company with which we partnered for this study, developed 31 different behavioural recommendations for the three severity levels. Depending of the geographical situation, the time of the year and other factors, the forecaster decides ad-hoc which behavioural recommendations to include in the message. Thus, we include a Table with a selection of only some of the recommendations.

"In Table 1 we list the general behavioural recommendations that were provided in both standard and impact-based warnings." Change Table numeration

**Table 1. Behavioural recommendations per severity level in the warning messages.** Note that this list is not exclusive.

| Warning severity level | | |
|---|---|---|
| *Moderate (level 1)* | *Severe (level 2)* | *Very severe (level 3)* |
| Don't make fire | Avoid wind-exposed areas | Be aware of falling objects |
| Close windows | Secure lose items | Follow instructions of emergency services |
| Drive slowly | Avoid forests | Seek protection in buildings |

Comment part 2: The term SW is chosen a little bit unfortunate, as in the Europeancontext standard meteorological warnings usually just include location, timing, hazard-type, severity level and eventually some meteorological information (e.g. Rainfall withamounts up to 100 mm), but generally no (generic) behavioral recommendations. Inthe Sendai context, behavioral recommendations are often seen, together with theimpact description, already as a part of an IbW or Impact-oriented warning (e.g. [1]).

Response: The Reviewer pointed out correctly that most European Meteorological Services issue standard warnings that do not include behavioural recommendations. However, we decided to keep the term SW as these are the standard warnings of our „weather partner" and are still close to the average standard warning in Europe. In the end, we believe that it is just a definition and with the additional Table, it should be clear what we understand under SW. However, we acknowledge the Reviewer's comment in a sentence and also include the suggested reference.

"It's important to note that most European Meteorological Services do not include generic behavioural recommendations in their standard warning *."

* Kaltenberger, R., Schaffhauser, A., and Staudinger, M.: "What the weather will do" – results of a survey on impact-oriented and impact-based warnings in European NMHSs, Adv. Sci. Res., 17, 29–38, https://doi.org/10.5194/asr-17-29-2020, 2020.

Comment part 3: Obviously some of the additional impact descriptions according to Table 1 (e.g. "Fallingof smaller branches") are very similar to the behavioral recommendation ("Präventions-Tipp") of the depicted SW in Fig. 1, saying to be aware of falling items ("Vorsicht vorherunterfallenden Gegenständen"). This provision of rather little additional informationin the warning message might be one of the main reasons for the principal finding, thatin their field experiment IBW did not result in greater behavioral response compared toSW which should be discussed by the authors. It is likely, that SW without behavioraladvices, IBWs with richer (or more empathic) impact descriptions or generally moretailormade warning texts would have changed the results significantly. Acknowledgingthis and other limitations, e.g. just warnings for one hazard were investigated, nowarnings with the highest severity level due to the relatively short period of just 2.5months, some concluding statements would benefit from being expressed a little bitmore cautious and less generalized.

Response: We thank the Reviewer for his comments regarding the discussion of the results. We included these in the conclusion section of the paper.

„Also, we should be cautious in generalizing the results as these are somehow contextually dependent. The provision of rather little additional information in the warning message might be another reason that in the field experiment IBW did not result in greater behavioural response compared to SW. It could be that SWs without behavioural recommendations, and IBWs with stronger language and richer impact descriptions could have resulted in different findings. "

In the final paragraph, we highlight some of the limitations of the study. We explain that the lack of very severe hazards may have influenced our results, as well as the fact that we only investigated the hazard wind. Based on the Reviewer's feedback, we also mentioned the relatively short study period that was in winter (and results may differ in summer).

„Thus, additional research could analyse whether these results are also valid for other natural hazards, as well as for different time periods in the year".

**Reviewer 2**

Introduction :

Comment #1: In the introduction, I appreciate that the authors try to draw on relevant theory from psychology and decision sciences. However, how the introduction stands now, it is long and in some parts confusing. I'd recommend to focus more on the applied value of this applied study and the factors varied in them, and what it adds to the current literature. The cited work of Casteel (2016, for example) or Andrea Taylor, University of Leeds, UK (2019) are excellent examples for how this could be done. If it is necessary to draw on several theories from cognitive and decision sciences, please clarify what are the precise predictions regarding the main independent variables explored here, based on these theories. This should be either based on theories, previous findings, or ideally integrate both. Regarding theory, I'd suggest narrow the theoretical focus and delineate precise predictions; and then pick these up again in the discussion and discuss whether and how they have been met, and why. If however, the aim of the authors is to review several theories rather than focusing on only one, please clarify why and how those were selected and relevant for IBW (for example construal level theory would then currently be missing).

Response: We are somewhat confused by this comment, as indeed we attempt to do in the introduction exactly what the reviewer suggest. In the first paragraph we focus on the question of whether more information leads to improved decisions, with a particular focus on the additional information contained in impact-based warnings (IBW) over standard warnings. Our question is thus whether it is worthwhile to include the added information that IBW contain. We then describe two different theoretical predictions, based on different cognitive pathways leading to human behaviour. When the analytic pathway is engaged, the rational actor model suggest that IBW will necessarily lead to improved decision-making. Where the affective pathway is engaged, it is unclear whether IBW will trigger improved decision-making. Additionally, we suggest that the method of investigating the response to IBW is important, because some methods (e.g. hypothetical surveys) make it more likely that the analytic pathway will be engaged, and these methods would then have an inherent bias in favour of showing IBWs to be superior. To capture the likely behaviour in real-world situations, we argue, it is necessary to replicate the conditions that could lead the affective pathway to be engaged. In this case, it is unclear whether IBWs will outperform SWs, and indeed the results may be highly context specific. Because we have tried to capture these points in the introduction, we are unclear how to revise it to meet the reviewer's request.

Comment #2: On a side note, I'd like to put forward that a dual systems-approach is widely ac- cepted is not correct – the 'ecological rationality' approach (see reference to Gigerenzer & Gaissmaier, 2011 below) is an entire interdisciplinary research paradigm in psychology and decision sciences (in social sciences more broadly, a much wider range of theories exist). It contrasts the dual system-approach. In my view, the inherent problem of the latter is that it doesn't clarify, generally and also in potentially threatening situations where people may rely on weather warnings, what a 'rational' and there- fore 'correct' decision actually is. Extreme weather leads to situations characterised by high uncertainty, where not all information is known and a 'rational' decision, based on all available information, is impossible to make. Also, quick and intuitive decisions could potentially be very helpful and adaptive in such uncertain emergency situations, if performed in the right decision context. While the introduction of the current paper is not the place to reflect this entire debate, the point I wish to illustrate is that the most important existing theories and predictions based on these need to be selected and described more carefully.

Response: We thank the reviewer for this side note, and agree that it does not require a change to the manuscript.

Comment #3: Having a more clear outline in the introduction would also clarify from an applied viewpoint what type of behavioural response is actually adequate in the authors' view (and 'rational', though I'd recommend to skip this term overall) – this may be based on general recommendations by weather warning services, or insurances, or Table 2? If there is an adequate response, please describe it in the introduction. If not, that also needs to be clear, because then there isn't such a thing as a rational decision in this context. A more applied focus as mentioned above would help in clarifying this.

Response: As with comment 1, we are challenged as to how to modify the manuscript to address this concern, as it is one that we have tried to address in our original submission. It is clear that in most situations, there is not one "correct" or "rational" response. In our study, our dependent variable is whether people attempt to change their behaviour in response to a piece of information. While we make no claim that a change of behaviour is a "correct" response for all people, it is also the case that a change in behaviour suggests that people both took note of the information and chose the change their behaviour, whereas not changing behaviour indicates either (a) that they failed to take note of the information, or (b) that they took note of it and decided actively not to change their behaviour. Observing a greater rate of behavioural change would suggest that at least some people who would have otherwise fallen into group (a) did take note of the information.

Comment #4: This would also help re-structuring the abstract according to short background, research question, design, method and sample, main findings and a short comment on results. Currently, it doesn't precisely reflect what was done here – and lacks for ex- ample the impact-based weather warning manipulation, or introduces the term 'stress' which doesn't re-appear prominently anywhere else (as far as I can see).

Response: We thank the reviewer for this comment, and agree that the abstract could be restructured to better reflect the overall content of the paper. The new abstract is as follows:

"When public agencies provide information provision to help people make better decisions, they often face the choice between parsimony and completeness. For weather services warning people of high-impact weather events, this choice is between offering standard warning (SWs) only of the weather event itself , such as wind-speed, or also describing the likely impacts (so called impact-based warnings, IBWs). Previous studies have shown IBWs to lead to a greater behavioural response. These studies, however, have relied on surveys describing hypothetical weather events; given that participants did not feel threatened, they may have been more likely to process the warning slowly and analytically, which could bias the results towards finding a greater response to the IBWs. In this study, we conducted a field experiment involving actual and potentially threatening weather events, where there was variance with respect to the time interval between the warning and the forecasted event, and where we randomly assigned participants to receive SWs or IBWs. We observe that shorter time intervals led to a greater behavioural response, suggesting that fear of an imminent threat to be an important factor motivating behaviour. We observe that IBWs did not lead to greater rates of behavioural change than SWs, suggesting that where fear is a driving factor, the additional information in IBWs may be of little importance. We note that our findings are highly contextualized, but we call into question the prevailing belief that IBWs and necessarily more helpful than SWs."

Methods:

Comment #5: #5 P.5 l.10 says "we asked participants whether the weather described inthe warning would pose a risk to them and whether it would affect them in carrying outtheir usual activities (e.g., commuting, working, shopping etc.). If they answered yes,they continued with the survey" – wouldn't this exclusion of people who don't change,be an explanation of why no effects were observed – if for example more people didn'treact to a SW, compared to a IBW?

Response: These questions that the Reviewer is referring too, helped to filter out those people who were not affected by the warning at all, because they were for instance the whole day at work and thus the wind warning from 2 to 5 pm would not be relevant for them at all. In the following we asked questions whether people changed behaviour as we also explain in the manuscript. So, there was a slight misunderstanding: we did not exclude people who did not react to the warning, but only those for which the warning was not relevant in the first place!

Comment #6: Page 5 l.34 reports that the regression isconducted on behaviour, but the regression table header indicates that it predicts be-haviour 'change' – please clarify whether it predicted absolute behaviors or a differ-ence score reflecting change (I don't think it did the latter).

Response: The Reviewer is right it is about this comment on behaviour and thus we adapted the language in the Table heading "…with behaviour as dependent variable".

Comment #7: Table 2: Please reporta measure of dispersion for age, and a range.

Response: We included a measure of dispersion (St. dev.) and a range

Comment #8: Figure 2: I think this is not aboutlikelihood (as the header implies) but a continuous score measuring actual behavior. Ithink the figure header should accordingly be something like "Mean self-reported be-haviour in response to two warning types,...". Please avoid the use of acronyms orexplain them again in the caption to allow the figure to stand on its' own. I'd also preferto see the full scale that was presented to participants on the y-axis; to get an ideaabout effect magnitude. Please do also apply these points to all other figures where appropriate.

Response: The Reviewer is absolutely right with respect to the measuring scale. We also avoid acronyms. Thus, we changed the header into "Mean self-reported behaviour in response to two warning types (standard and impact-based) and the three lead times (no, short and long), respectively the two severity levels (moderate and severe)." However, we decided to not present the full scale on the y-axis as this would deteriorate the readability of the figure. Also, the scale is explained in the caption.

Comment #9: Table 3 and overall results section: The use of standard significancelevels and *** has been widely criticized in psychology and decision sciences, andother fields (see for example https://www.nature.com/articles/d41586-019-00857-9 orhttps://journals.sagepub.com/doi/abs/10.1177/0959354302126005 for some context).Albeit still widely in use unfortunately, I'd thus refrain from marking results with *** andin bold, dependent on these levels; and follow reporting standards outlined in this litera-ture; including for example confidence intervals where appropriate for all study results.

Response: We thank the Reviewer very much for her comment. As the reviewer highlights, the use of standard significance levels is widely used in research, we believe that this approach is absolutely fine even though there exists some criticism. We also included error bars in the figures and indicate standardized coefficients in the regression analysis.

Discussion:

Comment #10: As the authors note in line 27 on page 7, the study is based on self-reports and a self-selected sample. Please acknowledge the dearth of literature on samples and self-reports in psychology and social sciences more generally where this has been criticized, and provide a recommendation for future studies.

Response: We are happy to include this note. We have revised the relevant text as follows, following the reviewer's suggestion:

"This may indicate higher levels of weather awareness and knowledge, which could also be another explanation for the lack of effect of warning type. There is a dearth of literature on the effects of such self-selection in social science research, though ideally researchers would design field experiments where self-selection is not present."

Technical corrections

Comments #11 - #17: Were all addressed. We only did not include the comparison to the general population in the table (comment #15) as the table is explicitly about the results of the study. However, the information is still available in the text.

[revised manuscript text omitted]